# Performance Enhancement in Powder-Fabricated Cu_2_(ZnSn)Se_4_ Solar Cell by Roll Compression

**DOI:** 10.3390/ma16031076

**Published:** 2023-01-26

**Authors:** Jaehyun Park, Hyobin Nam, Bong-Geun Song, Darya Burak, Ho Seong Jang, Seung Yong Lee, So-Hye Cho, Jong-Ku Park

**Affiliations:** 1Materials Architecturing Research Center, Korea Institute of Science and Technology, Seoul 02792, Republic of Korea; 2Division of Nanoscience and Technology, KIST School, Korea University of Science and Technology, Seoul 02792, Republic of Korea

**Keywords:** Cu_1.7_Zn_1.2_Sn_1.0_S_4.0_ (CZTS), Cu_2_ZnSnSe_4_ (CZTSe), light absorbing layer (or film), thin film solar cell, powder process, roll compression, densification, selenization

## Abstract

Despite the improved conversion efficiency of Cu_2_(ZnSn)Se_4_ (CZTSe) solar cells, their roll-to-roll fabrication nonetheless leads to low performance. The selenization time and temperature are typically considered major parameters for a powder-based CZTSe film; meanwhile, the importance of the densification during the roll-to-roll process is often overlooked. The densification process is related to the porosity of the light-absorbing layer, where high porosity lowers cell performance. In this study, we fabricated a dense CZTSe absorber layer as a method of controlling the compression of a powder precursor (Cu_1.7_(Zn_1.2_Sn_1.0_)S_4.0_ (CZTS)) during the roll-press process. The increased particle packing density of the CZTS layer was crucial in sintering the powder layer into a dense film and preventing severe selenization of the Mo back electrode. The pressed absorber layer of the CZTSe solar cell exhibited a more uniform chemical composition determined using dynamic secondary ion mass spectrometry (SIMS). Under the AM 1.5G illumination condition, the power conversion efficiency of the pressed solar cell was 6.82%, while the unpressed one was 4.90%.

## 1. Introduction

Cu_2_ZnSnSe_4_ (CZTSe) has drawn significant attention as a light-absorbing material for thin film solar cells owing to its elemental abundance compared to Cu(InGa)Se_2_ (CIGS) [1]. Having many similarities with CIGS, CZTSe films are generally prepared using fabrication methods developed for CIGS solar cells without much modification [1,2,3,4,5]. For example, a powder process which includes coating a powder layer and its sintering into a thin film was adopted to minimize the material loss in CIGS films and has been successfully applied to CZTSe films [6,7]. Although the powder process has the considerable advantages of high material usability and low facility installation costs, it is yet to be applied to CZTSe solar cells on a larger scale. A high-density CZTSe film requires a thin layer, typically in the range of 1 to 3 µm, which is difficult to obtain using the powder process. Moreover, applying high temperatures to achieve adequate sintering of the powder layer is detrimental not only to a glass substrate but also to the compositional stability of CZTSe [1,8,9].

The maximum power conversion efficiency of the powder-processed CZTSe solar cell reported to date was 10.1% in 2012 [10], where a hydrazine-based solution technology was used [11]. Until recently, the best efficiencies of CZTSe solar cells have been achieved using a vacuum-processed CZTSe layer with hydrazine [12]. However, the performance of these CZTSe solar cells is limited by defects in the CZTSe light absorption layer, which prompted research into defect-related loss mechanisms and alternative routes for CZTSe solar cell fabrication [13,14]. The high-density CZTSe film fabricated by the powder process is a promising candidate to lower defect density and increase cell efficiency. Moreover, the roll-to-roll device fabrication is a potent process for the mass production of powder-based CZTSe. For example, Kurley et al. reported a conversion efficiency of 10.4% for a CdTe solar cell using the roll-to-roll technique [15]. However, successful results have yet to be reported for CZTSe fabrication based on the roll-to-roll process [16]. In particular, research into the powder-based CZTSe solar cell implementation on a flexible substrate is limited and scarce. The highest conversion efficiency of CZTSe solar cells on a flexible substrate reported to date is 6.1% via sequential sputtering on stainless steel [17].

Compression is an integral part of the roll-to-roll process. Chen et al. fabricated a smooth and dense CZTSe film using compression but did not report power conversion efficiency [18,19]. Particle size and packing density are essential in the sintering process of particles. A comparatively lower temperature is required for densification because the packing density is higher and the particle size is smaller [17]. Therefore, the powder process requires fine particles and high-density packing to fabricate a dense CZTSe film.

Another effective way to provide a dense CZTSe film is selenization. The selenization process involves treating sulfur-containing materials (e.g., CZTS) with selenium vapor at high temperatures so that the sulfur is substituted for the selenium (e.g., CZTSe) [1,10]. This elemental substitution process reduces the porosity of CZTSe thin films by cell volume expansion attributed to the different ionic sizes of Se and S (184 and 170 pm, respectively) [20,21].

However, if controlled, the selenization process for the CZTSe solar cells can lead to detrimental effects, such as compositional changes of an absorber layer, degraded back electrodes (Mo), and poor interfacial contacts. For example, Taskesen et al. fabricated a CZTSe solar cell through the selenization of a CZT layer [22]. In this cell, the CZTSe absorber layer grew 6-fold thicker than the pristine CZT layer through severe selenization, which resulted in a rough surface, damage to the Mo back electrode, and poor interfacial contacts between the Mo and CZTSe.

The present study provides a method to increase the particle packing density in the Cu_1.7_(Zn_1.2_Sn_1.0_)S_4.0_ (CZTS) powder coatings that are printed with CZTS fine particles produced by mechanochemical ball milling [4]. The densification of the CZTSe layer was controlled by compressing the CZTS film using a commercial roll-press machine and post-annealing at a relatively low temperature. In addition, selenium vapor was supplied during the heating to provide further grain growth and densification by transforming CZTS to CZTSe. Furthermore, the selenization rate was controlled by the compression, which limited the elemental loss of the CZTSe layer and maintained intact contact between the CZTSe and Mo. The present approach is not limited to the fabrication of the CZTSe thin film and can be generally applied to thin-film sintering using the powder process.

## 2. Materials and Methods

CZTS powder was synthesized using a mechanochemical method with elemental precursors of Cu, Zn, Sn, and S. First, the CZTS powder was mixed with anhydrous ethanol and ball-milled for 72 h to create a paste with a 20% solid coating. Then, the paste was coated on a substrate using the doctor-blade method and dried. For the densification process, a roll-press machine (WCRP-1015G, Wellcos Corp., Rep. Korea) was used to pass the dried CZTS coating through a pre-gap (resolution: 1 μm) at a speed of 0.2 m/s without heating. The roll compression of the CZTS coating was performed by setting the point where the substrate was not broken to 0 and changing it to the desired thickness (Figure 1).

Next, heating was performed at a rate of 2 °C/min to 570 °C and maintained at 570 °C for 30 min in a 2% H_2_/N_2_ gas environment. Se vapor was supplied during the heating by placing a mixture of Se and Al_2_O_3_ powder in the same chamber for the selenization of CZTS [9]. A solar cell had a device structure of soda lime glass (SLG)/Mo/CZTSe/CdS/i-ZnO/ZnO:Al/Ni/Al in a sequence. A Mo back electrode layer (~500 nm) was sputtered, and a CdS layer (50 nm) was deposited on the CZTSe film using a chemical bath method. i-ZnO and ZnO:Al layers (50 and 600 nm, respectively) were sputtered, and Ni/Al grid (50 and 500 nm) was thermally evaporated.

Scanning electron microscopy (SEM) images were obtained using Inspect F (FEI, USA) scanning electron microscope with an acceleration voltage of 15 kV, and X-ray diffraction patterns were obtained using D8 Advanced (Brucker Corporation, USA) diffractometer. The depth profiles of unpressed and pressed ZnO/CdS/CZTSe/Mo were analyzed using dynamic secondary ion mass spectrometry (SIMS, IMS 4FE7, Cameca) with a Cs^+^ ion gun (impact energy of 5.5 keV). Optical bandgaps were measured using UV/Vis transmission spectroscope (Cary 5000). After calibration, the current−voltage (j−V) curves were obtained using a class-AAA solar simulator (Yamashita Denso, YSS-50S). After calibration, external quantum efficiencies (EQEs) were measured using an incident photon-to-current conversion efficiency measurement system (PV Measurements, Inc., Boulder, CO, USA).

## 3. Results and Discussion

A suspension of the mechanochemically synthesized Cu_1.7_Zn_1.2_Sn_1.0_S_4.0_ powder and ethanol in a 0.16 g/mL concentration (Figure 2a insets) was used to prepare the CZTS powder coatings using the doctor-blade method. First, a condition where the amount of the dry CZTS powder coating equaled the weight of a bulk CZTS film with a thickness of 2 μm was determined through doctor-blading. Then, after drying overnight, the powder coating was subjected to a roll press to increase its particle packing density. Figure 2a,d represents the macroscopic features of the unpressed and pressed powder coatings on the SLG substrates. The unpressed coatings exhibited a bluish-dark gray color, while their pressed counterparts appeared as a black film with a metallic luster. This observation indicated that the roll press was successful as the particles on the surface of the pressed powder coatings were packed closely enough to reflect light [12]. These results can be further confirmed by comparing the top-view SEM images shown in Figure 2b,e.

Noticeable pores can be observed in the unpressed CZTS layer without an additional densification process. In contrast, the CZTS layer formed by the densification process resulted in reduced and planarized pores. Furthermore, as shown in Figure 2c,f, the cross-sectional SEM images of each powder coating revealed that the densification by the roll press contributed to a film thickness reduction by ~29% (from 5.1 to 3.6 μm) and resulted in a highly dense and flat top surface. The porosity of a powder coat was calculated as a volume fraction of the void space in the powder coats. First, the weight of coated CZTS was measured by comparing weights before and after powder coating and oven-drying, which was then converted to volume using the density of bulk CZTS of 4.56 g/cm^3^. This value was assumed to be an ideal volume of a dense CZTS film without any void space. Next, measured volumes of the unpressed and pressed CZTS coats were calculated based on the coated area and the film thicknesses obtained using SEM images, which were then used to obtain volumes of the void space in each coat by subtracting the ideal volume from the measured one. As a result, the porosity of the unpressed coat was 60.1%, and the pressed one was 44.4%. The lower porosity of the pressed one should thus lead to sintering under more moderate conditions [23].

The CZTS sintering processes reported to date indicate that unpressed and pressed coatings are barely sintered under the inert atmosphere at temperatures of 500~600 °C, which is much lower than the melting temperature of Cu_2_ZnSnS_2_ (990 °C) [24]. Consequently, the transport of each element of CZTS within the coating is limited at low temperatures; thus, using sintering aids was necessary to promote mass transport. For CZTS (or CIGS) light-absorbing layers, Se or S vapor is often used as the sintering aid, which is known to induce the liquid phase-assisted transport of CZTS elements within the layer [25,26].

The density of the porous CZTS coating was increased by controlling the pre-gap of the roll-press machine. Figure 3 demonstrates the correlation between film thickness and the degree of compression. Figure 3a,b contains cross-sectional SEM images of CZTS thin films coated by the same doctor-blade method but pressed by different pre-gap settings of 4 and 2 μm, respectively, which are as thick as 4.7 and 2.1 μm, respectively. During this process, noticeable damage to the Mo back electrode layer of 0.5 μm in thickness was found. For selenization, the pressed CZTS layer was heat-treated at 570 °C for 30 min in the reductive atmosphere (2% H_2_/N_2_ gas). The resulting CZTSe layers were 4.2 and 2.4 μm in thickness, as shown in Figure 3c,d, respectively, indicating that the film with higher porosity exhibits volume reduction. In comparison, the one with lower porosity undergoes volume expansion. The thickness correlation of CZTS and CZTSe layers to pre-gaps of the roll-press machine is drawn in Figure 3e, further confirming the trend of volume reduction with the pre-gap setting of 4 or higher and slight volume expansion with the pre-gap setting 3 or lower. In the meantime, the Mo electrodes of both CZTSe layers in Figure 3c,d show a slight increase in their thickness (0.5 to 0.7 μm), indicating the partial selenization of Mo electrodes to MoSe_2_.

On the other hand, the porosity of the formed CZTSe layer changed greatly. It can be seen that the porosity of the CZTSe was significantly reduced with the increase in pressure. These trends were the same for the CZTSe layers and the MoSe_2_ electrodes. The thickness of the 2.4 μm CZTSe light absorber layer shown in Figure 3d was larger than the formed grain size. The grain boundary, a representative factor that lowers the performance of the CZTSe solar cell, leads to a higher recombination rate and decreased carrier separation [13]. In order to shape the CZTSe grains formed after selenization into a single layer, the experiment was conducted by reducing the amount of the CZTS coating and the pre-gap of the roll press to 1 μm. An unpressed sample was also manufactured to compare the effect of roll compression. After selenization, the CdS/i-ZnO/ZnO:Al/Ni/Al layers were sequentially deposited to complete the devices.

Figure 4a,b demonstrates the top-view SEM images of unpressed and pressed solar cells, respectively, showing reduced porosity and roughness for the pressed coating. The cross-sectional SEM images of the unpressed and pressed absorber layers with a device structure of SLG/Mo/CZTSe/CdS/i-ZnO/ZnO:Al/Ni/Al are shown in Figure 4c,d. The two absorber layers exhibited different film morphology, with the pressed coating showing a thinner layer (~1.3 μm) with larger grains. In particular, in the light absorber layer of the pressed cell, many single grains were directly connected between the Mo layer and the CdS layer. At the same time, the unpressed counterpart was thicker (~1.8 μm) with smaller grains and larger volumes of pores, which, as a result, allowed Se vapor to permeate and react with Mo in the back electrode, forming a thick MoSe_2_ layer with a thickness of approximately 1.0 μm, as shown in Figure 4c. On the other hand, the Se vapor’s transfer to the Mo electrode was suppressed in the pressed coating due to its increased density, which resulted in a MoSe_2_ layer with a lower thickness of ~0.7 μm, as shown in Figure 4d.

Depth profiling of the device structures, ZnO/CdS/CZTSe/Mo, with the unpressed and the pressed absorber layer was performed with dynamic SIMS to examine their composition difference (Figure 5). For a fair comparison, the elemental profiles of Cu and Sn were normalized to Se as Se should be equally present owing to the same selenization process. In addition, the Zn profile was excluded for simplicity because it appears in both the transparent electrode (ZnS) and absorber layer (CZTSe).

When comparing the amounts of Cu and Sn in the CZTSe layer, the unpressed sample showed ~0.90% loss of Cu and ~0.36% loss of Sn relative to those of the pressed sample, as shown in Figure 5. In particular, the initial hills of the Cu and Sn profiles in the unpressed CZTS layer suggest that the loss of Cu and Sn was more severe on the bottom layer than the top owing to the elements’ diffusion from the bottom to the top through the developed pores and subsequent condensation on the top layer. These phenomena are caused by the liquid-assisted sintering of Se. In our selenization process (Se is supplied as a solid at 570 °C), selenium undergoes melting (melting point: 221 °C) and subsequent vaporization (vapor pressure: 10 kPa at 540 °C) [27]. Se vapor then condenses on the surface of the CZTS film as a liquid and contributes to the substitution of S to Se and the resulting CZTSe grain growth by liquid-assisted sintering [28]. As the unpressed CZTS powder coat has many open pores on its top surface and a large void space (Figure 2b and Figure 4a), more Se vapor can condense in the coat, compared to the pressed coat. Although the excessive Se is removed by continuous heating during the selenization, elemental dissociation of CZTSe may result in the compositional imbalance, evidenced by the depth profile. CZTSe is known to thermally degrade to binary and ternary compounds such as Cu_x_Se_y_ and Cu_x_Sn_y_Se_z_ [29]. Thus, it is considered that in the excessive Se supply, the partial dissociation of CZTSe to those compounds was accelerated, which resulted in the uneven distribution of Cu and Sn in the unpressed coat. The compositional imbalance of the unpressed CZTSe cell is a major reason for the low cell efficiency.

The depth profiles of Mo in the back electrodes of the unpressed and pressed devices show that the diffusion of Se to the Mo electrode occurred in both the unpressed and pressed devices. However, in the pressed one, the Se diffusion to the Mo electrode forming MoSe_2_ occurred within the shallow top layer of the Mo electrode (~0.3 μm). In contrast, in the unpressed one, Se was found to diffuse into the entire Mo back electrode, which is consistent with the SEM observation indicating a larger thickness of the Mo back electrode of the unpressed device than one of the pressed.

The XRD patterns of the CZTSe layer formed on the Mo back electrode with and without compression, as compared in Figure 6a, indicated that all of the S of CZTS is fully substituted to Se in the selenization process. The XRD analysis also revealed that the crystallinity of the pressed absorber layer was higher than the unpressed layer. This higher crystallinity of CZTSe in the pressed absorber layer than in the unpressed one is attributed to its larger grain size, which originates from densification. MoSe_2_ crystals were observed in the pressed and unpressed samples (green diamond labels). In contrast, a Mo crystal peak only appeared in the pressed sample, indicating that the Mo electrode was fully transformed to MoSe_2_ in the unpressed coating. These results are in good agreement with those of the dynamic SIMS analysis. The magnified XRD patterns shown in Figure 6b further compared the crystallinity of the MoSe_2_ layers of the pressed and unpressed samples. The higher MoSe_2_ peak of the unpressed sample indicated severe selenization of the Mo electrode compared to the pressed one. The excessive selenization of Mo should be avoided as it causes delamination of the back electrode from a substrate due to the significant volume expansion by the transformation from Mo to MoSe_2_ (almost four times) [30,31]. In contrast, the formation of a thin layer of MoSe_2_ should have a beneficial effect on the adhesion of the CZTSe absorber to the Mo back electrode and the semiconductor/metal contact, which changes from a Schottky (CZTSe-Mo) to Ohmic type (CZTSe-MoSe_2_-Mo), thereby reducing the barrier of the hole transfer [32,33].

The photovoltaic characteristics of the solar cells were studied using the current density-voltage (*j*-V) curves (Figure 7a) and external quantum efficiency (EQE) measurements at each wavelength of incident light (Figure 7b) without a bias voltage. The solar cell device with the pressed CZTSe layer exhibited the maximum power conversion efficiency of 6.82% based on the active area (0.44 cm^2^), with a short-circuit current density (*J_sc_*) of 39.4 mA/cm^2^, an open-circuit voltage (*V_OC_*) of 0.372 V, and a fill factor (FF) of 46.5% (red lines in Figure 7a, see Appendix A for raw data). On the other hand, the unpressed CZTSe layer showed the maximum power conversion efficiency of 4.90%, with a *J_sc_* of 30.1 mA/cm^2^, a *V_OC_* of 0.388 V, and FF of 41.9% (black lines in Figure 7a, see Appendix A for raw data). The power conversion efficiency was improved by almost 38.9%, and both *J_sc_* and FF increased considerably by 30.7 and 10.0%, respectively. The EQEs of the CZTSe solar cells, unpressed (grey line, *J_SC_* = 30.14 mA/cm^2^) and pressed with different degrees of densification (reddish lines, *J_SC_* = 33.03, 35.86, and 39.39 mA/cm^2^), were depicted in Figure 7b. As pressure was applied, *J_SC_* and EQE increased concomitantly. The EQEs of the pressed cell appeared considerably higher than the unpressed one in two different regions of wavelengths―400 ~ 500 nm and 500~1100 nm. The highest EQE of the pressed cell was higher by 12% than the unpressed cell (from 46.4 to 51.8%) at 450 nm, which contributes the most to the solar cell performance as the energy density is the highest in this region. Unexpectedly, the slightly pressed cell (Jsc = 33.03) showed a lower EQE at 450 nm than the unpressed one, which indicates that the insufficient densification detrimentally affected the dissociation of the CZTSe layer, leading to unknown impurity compositions. In a broad region of 500~1100 nm, the highest EQE of the pressed cell exhibited an increase of 17% (from 75 to 87.5% at 580 nm), compared to the unpressed cell. However, in the region above 1100 nm, the EQE of the unpressed cell demonstrated a rather broader response than the pressed cells, which is related to the bandgap of the absorber layer.

The bandgaps (*E_g_*) of all of the CZTSe absorber layers were obtained using diffuse reflectance UV–Vis spectra (DRS). The bandgap of the pressed layers was found at approximately 1.0 eV (Figure 7c), which is in a good agreement with the literature value of CZTSe [9,34,35]. However, the bandgap of the unpressed CZTSe film appeared at 0.9 eV (grey line in Figure 7c). As indicated by the SIMS analysis in Figure 5, the unpressed CZTSe layer exhibits a significant loss of Cu and Sn elements during sintering. Therefore, the lower bandgap of the unpressed absorber layer results from its inhomogeneous compositions. The pink lines in Figure 7b,c represent the EQE and bandgap of the pressed CZTSe cell that showed the lowest power conversion efficiency (5.884%) among the pressed cells. It is considered that its higher bandgap by approx. 0.5 eV than other pressed cells may result from insufficient selenization, causing the narrower absorption of light [35].

## 4. Discussion and Conclusions

The present study provides a facile and robust solution for preparing light-absorbing layers of CZTSe using powder processing technology. The porosity of the CZTS powder layer was lowered from ~60.1 to ~44.4% via roll pressing, which induced a higher particle packing density. The negative impact of a large volume change induced by selenization, an essential process for fabricating CZTSe solar cells from CZTS, was successfully minimized by fine gap tuning of the roll press. The large grains of CZTSe, formed by compression, were found to influence the efficient transfer of photo-generated charges to the Mo back electrode and the CdS buffer layer. The dynamic SIMS elemental depth profile showed that the CZTSe layer of the pressed cell had a uniform chemical composition compared to the unpressed cell. The densification process also prevented excessive selenization of the Mo back electrode. These features decreased the defect formation in the CZTSe light absorber layer and subsequently improved the contact property with the Mo back contact.

Cells based on the unpressed and pressed absorber layers were fabricated to confirm the improvement of the photovoltaic performance via the roll-press process. It was thus validated that the cell with the pressed absorber layer showed a 38.9% enhancement in maximum power conversion efficiency compared to the unpressed cell. These results demonstrate that the doctor-blading technique does not provide sufficient density for the powder-based absorber layer and needs further treatment, which was compression in this study. The roll press is a highly suitable method for mass production as the roll-to-roll process is universally applicable.

However, the powder-coated solar cell compression should be controlled because it can cause built-in stresses at the absorber layer/back electrode and selenized back electrode/metal electrode interfaces. Although not included in this study, no photovoltaic efficiency was observed in the case of the cells with compression of more than half the thickness of the original powder coat. Moreover, due to the volume expansion by substitution of the S with Se ions, the highly pressed film undergoes large changes in its volume, producing stresses at the interfaces of CZTSe/MoSe_2_ and the MoSe_2_/Mo back electrode. Such built-in stress results in delamination and poor contact between the layers. Therefore, precise compression control needs to be further implemented for this powder process to be industrially applicable.

## Figures and Tables

**Figure 1 materials-16-01076-f001:**
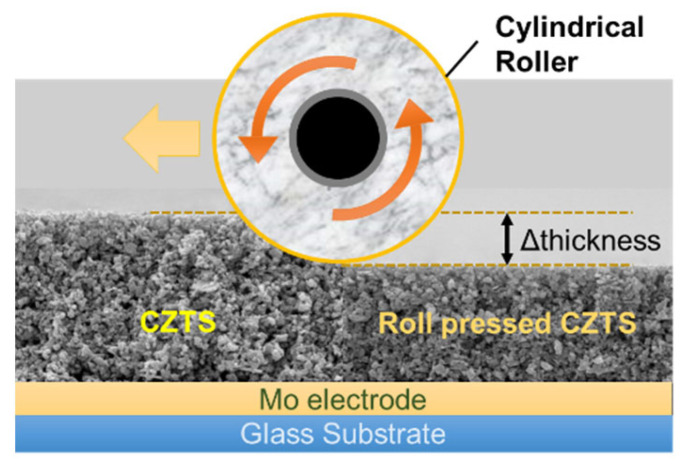
Compression process of the CZTS powder coat using a roll-press machine (Wellcos Corp., Rep Korea).

**Figure 2 materials-16-01076-f002:**
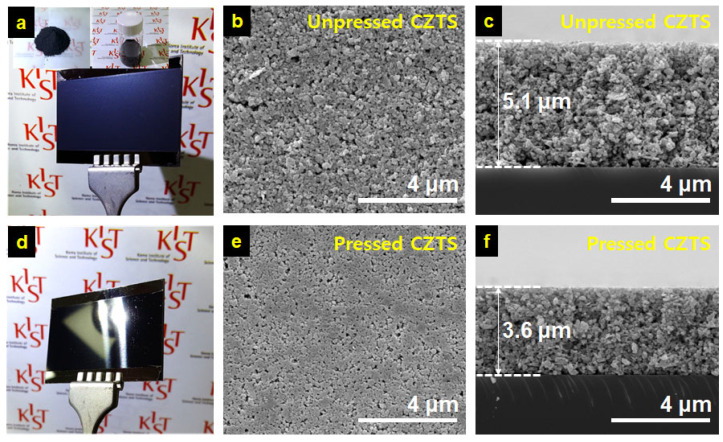
Photographic, top view, and cross-sectional SEM images of (**a**–**c**) unpressed and (**d**–**f**) pressed CZTS coats on SLG. Insets in (**a**) represent photographic images of the CZTS powder and its ethanol suspension from left to right.

**Figure 3 materials-16-01076-f003:**
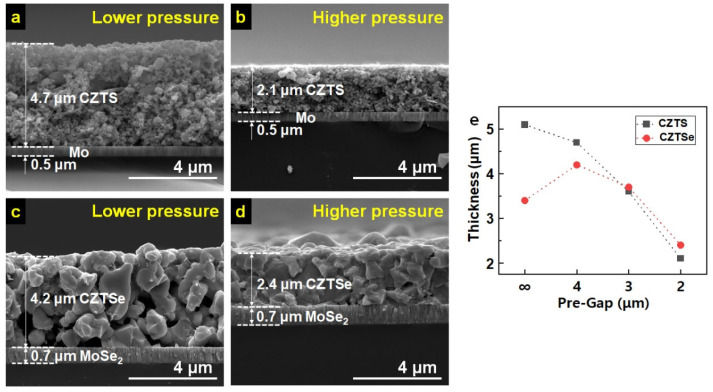
Cross-sectional SEM images of roll-pressed CZTS coats with (**a**) lower pressure and (**b**) higher pressure, respectively, on Mo/SLG. (**c**,**d**) Cross-sectional SEM images of the selenized CZTS (CZTSe) layers of (**a**) and (**b**), respectively. (**e**) CZTS thickness before and after selenization according to pre-gap of the roll press. (Black squares represent before selenization, and red circles represent after selenization.)

**Figure 4 materials-16-01076-f004:**
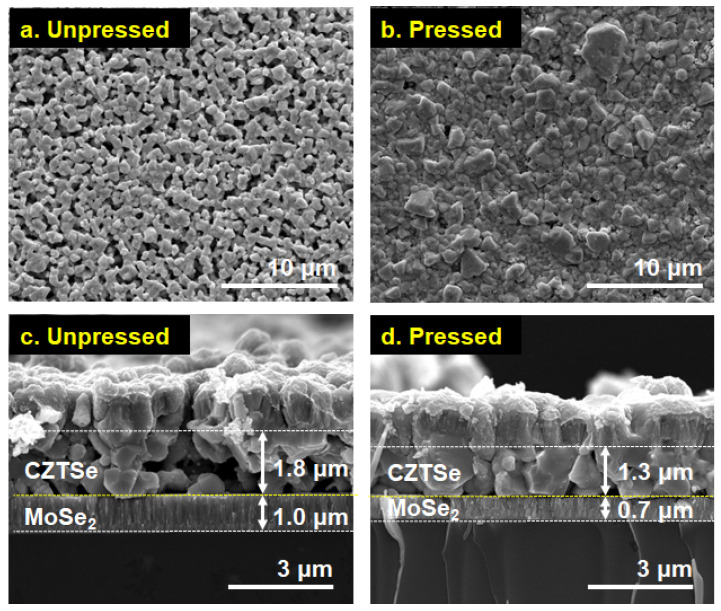
Top view and cross-sectional SEM images of CZTSe solar cells with (**a**,**c**) unpressed and (**b**,**d**) pressed absorber layer with a device structure of SLG/Mo/CZTSe/CdS/i-ZnO/ZnO:Al/Ni/Al. (Arrow bars indicate the thickness of each CZTSe absorber layer and MoSe_2_ layer.)

**Figure 5 materials-16-01076-f005:**
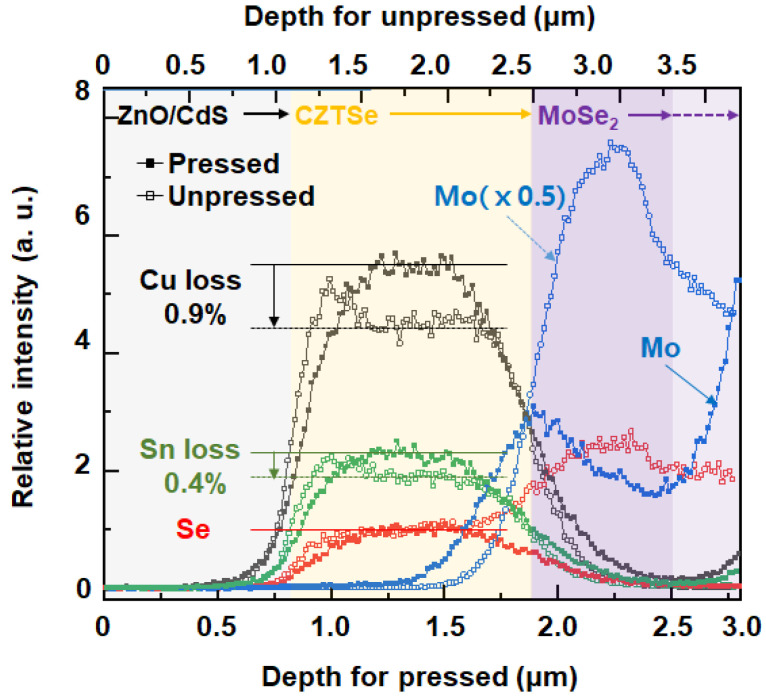
Dynamic SIMS elemental profiles of CZTSe/MoSe_2_ layer from the unpressed (unfilled squares) and the pressed (filled squares) devices. For comparison, the CZTSe thickness was normalized.

**Figure 6 materials-16-01076-f006:**
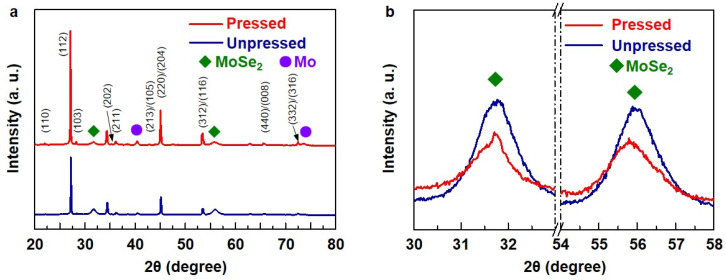
(**a**) XRD patterns of CZTSe solar cells with unpressed (bottom) and pressed absorber layers (top) (green diamond labels indicate diffraction peaks of MoSe_2_, and purple circle labels indicate those of Mo) and (**b**) magnified XRD patterns of MoSe_2_ (blue line represents the unpressed cell, and red line represents the pressed cell).

**Figure 7 materials-16-01076-f007:**
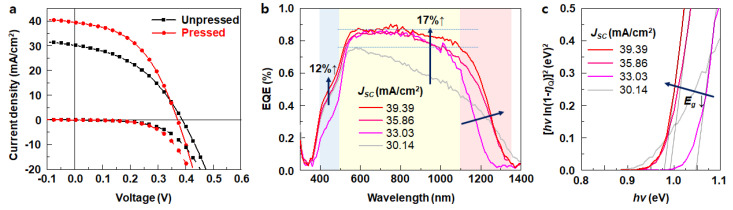
Photovoltaic performance of CZTSe thin film solar cells: (**a**) *j-V* curves under AM 1.5G illumination and in the dark; (**b**) external quantum efficiency (EQE) spectra; and (**c**) bandgaps of CZTSe thin film solar cells without a bias light under short-circuit conditions of unpressed (grey line) and pressed cells (reddish lines).

## Data Availability

Data are contained within the article.

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
