# Peer review of "Performance Enhancement in Powder-Fabricated Cu2(ZnSn)Se4 Solar Cell by Roll Compression"

_materials, 2023, doi:10.3390/ma16031076_

Round 1
Reviewer 1 Report
The article is about performance enhancement in powder-fabricated Cu2(ZnSn)Se4 solar cell by compression. However, some issues must to be addressed:
- Abstract: Please start by expressing the aim of this paper, followed by the rest of the information. Also, please define or try to avoid using abbreviations in the abstract. Typically, the abstract should provide a broad overview of the entire project, summarize the results, and present the implications of the research or what it adds to its field.
- The bibliographic foundation is important and well executed, however some new discussions should be inserted, authors should consider some works in the literature, such as: DOI 10.30638/eemj.2009.043.
- Figure 2: split in two different figures and comment separately.
- The results are merely presented, not properly discussed. Please add explanations for the observed changes. Please give an extended discussion on the obtained results and correlate your findings with previous literature studies and prospective applications.
- More analysis and interpretation of the results should be added for a clearer understanding of observed experimental phenomena.
- The authors must to provide some details about importance of the research and their applicability.
- Please rewrite the conclusions in a more quantitative form and enhance the clarity of the conclusion section in order to highlight the results obtained.
- General check-up and correction of the English language is suggested. There are still some minor typos and grammatical errors.
The author needs to address the abovementioned points for the betterment of the manuscript.
Author Response
We thank reviewer 1 for his or her reasonable comments. Please see the attachment for the answers to the comments.

Author Response
We thank reviewer 2 for his or her reasonable comments. Please see the attachment for answers to the comments.

Round 2
Reviewer 1 Report
There are too many self-citations of the coauthors.
Author Response
We agree with the reviewer's comments. We have, accordingly, revised the overall usage of the English language throughout the manuscript to provide a better understanding for future readers. We have also reduced the number of self-citations and substituted them with additional relevant studies in the solar cells field. We appreciate the time and effort the reviewer has dedicated to providing feedback and hope the implemented changes will be sufficient.
Reviewer 2 Report
Thanks the author for addressing all comments and modification.
Author Response
We agree with the reviewer's assessment. Accordingly, throughout the manuscript, we have revised the overall English language and implemented style corrections to increase the understanding of future readers. We sincerely thank the reviewer for their comments.